# DCT Underwater Image Enhancement Based on Attenuation Analysis

**DOI:** 10.3390/s25237192

**Published:** 2025-11-25

**Authors:** Leyuan Wang, Miao Yang, Can Pan, Jiaju Tao

**Affiliations:** School of Electronic Engineering, Jiangsu Ocean University, Lianyungang 222000, China; wly1515273@163.com (L.W.); wanying_pc@163.com (C.P.); njfu071tao@163.com (J.T.)

**Keywords:** underwater image enhancement, multi-channel attenuation analysis, contrast enhancement, DCT enhancement

## Abstract

Underwater images often suffer from color distortion, reduced contrast, and blurred details due to the selective absorption and scattering of light by water, which limits the performance of underwater visual tasks. To address these issues, this paper proposes an underwater image enhancement method that integrates multi-channel attenuation analysis and discrete cosine transform (DCT). First, the color statistics of an in situ-captured underwater image are mapped to those of a reference image selected from a well-illuminated natural image dataset with standard color distribution; no pristine underwater image is required. This mapping yields a color transfer image, i.e., an intermediate color-corrected result obtained via statistical matching. Subsequently, this image is fused with an attenuation weight map and the original input to produce the final color-corrected result. Secondly, taking advantage of the median’s resistance to extreme value interference and the Sigmoid function’s flexible control of gray-scale transformation, the gray-scale range is adjusted in different regions through nonlinear mapping to achieve global contrast balance. Finally, considering the visual system’s sensitivity to high-frequency details, a saliency map is extracted using Gabor filtering, and the frequency characteristics are analyzed through block DCT transformation. Adaptive gain is applied to high-frequency details to enhance them. Experiments were conducted on the UIEB, EUVP, and LSUI datasets and compared with existing methods. Through qualitative and quantitative analysis, it was verified that the proposed algorithm not only effectively enhances underwater images but also significantly improves image clarity.

## 1. Introduction

In many activities such as underwater welding, seabed exploration, scientific research, and biodiversity studies, high-quality visual information is a key foundation for supporting the execution of tasks [1]. However, the complex underwater environment poses significant challenges regarding obtaining clear images. The selective attenuation of light by water and the scattering and blocking effects of suspended particles result in common problems in underwater images, such as low visibility, color distortion, insufficient contrast, and blurred details [2]. Although many underwater image enhancement methods have been proposed, most of them fail to achieve good detail enhancement.

At present, underwater image enhancement technologies are mainly classified into three major categories: non-physical model methods, physical model-based methods, and deep learning-based methods. Non-physical model methods [3,4,5] take image processing technology as the core and do not rely on complex optical physical modeling. They improve image quality through classic means such as histogram equalization, contrast enhancement, denoising and smoothing, super-resolution reconstruction, and enhancement filters. However, the variability of image degradation makes it difficult for a single image processing operation to adapt to all scenarios. On the other hand, some methods may cause image information loss or distortion during multi-domain transformation. Therefore, it is necessary to comprehensively consider the balance of image quality in multiple feature dimensions.

In recent years, underwater image enhancement techniques based on physical models [6,7,8] and deep learning [9,10,11] have seen significant progress. Due to the difficulty in comprehensively covering all underwater optical effects, physical model methods [12] have limited generalization ability across different scenarios. Meanwhile, deep learning-based methods [13] are constrained by the difficulty in obtaining high-quality paired images, resulting in a deviation between the model training data and the real underwater environment. Additionally, they suffer from weak interpretability, high computational resource consumption, and complex parameter tuning.

This paper introduces a novel method for enhancing underwater images by integrating multi-channel attenuation analysis with discrete cosine transform (DCT). The approach employs a collaborative optimization strategy for color correction, contrast enhancement, and detail enhancement, resulting in a comprehensive improvement in underwater image quality. In particular, the color correction module generates a dynamic weight map that combines the attenuation indices of the red and blue channels. This adaptive mechanism effectively compensates for color loss in various regions of the image. The contrast enhancement module utilizes a mean–median fusion threshold segmentation histogram and applies Sigmoid nonlinear mapping to separately enhance the background and foreground regions. Additionally, the detail enhancement module employs Gabor filtering to extract a visual saliency map and establishes a three-level gain mechanism based on the AC energy and frequency characteristics in the DCT domain. This mechanism enhances details in salient regions while mitigating blocking artifacts through a weighted superposition strategy. The key contributions of this study are outlined as follows:(1)A color correction algorithm based on multi-channel attenuation analysis is proposed. By combining CIELab space-based color statistical matching with red/blue channel attenuation/scattering physical models, it achieves adaptive color distortion compensation in complex scenarios like strong and weak light.(2)A contrast enhancement strategy based on mean–median fusion segmentation is designed. Combined with Sigmoid nonlinear mapping, it suppresses extreme value interference while precisely regulating the gray dynamic range, effectively solving the contrast imbalance.(3)A DCT detail enhancement mechanism is constructed. By guiding DCT domain frequency gain adjustment with visual saliency, it enhances edge and texture details while effectively avoiding distortion caused by amplification of high-frequency noise and excessive sharpening of details, achieving a balance between enhancement effect and image naturalness.

## 2. Related Work

### 2.1. Underwater Image Enhancement Method Based on Physical Model

Underwater image enhancement methods based on physical models rely on Jaffe’s foundational imaging model [14]—which decomposes received light into three physical components—and reconstruct clear images by quantifying the optical parameters of water. Existing methods mainly fall into three categories: first, those centered on He et al.’s Dark Channel Prior (DCP) [15], including Wang et al.’s [16] extension to more scenarios via adaptive attenuation curves and Ravi et al.’s [17] optimization of color cast and brightness by combining underexposure correction. Second, there are methods focused on multi-function integration, such as Li et al.’s [18] method that integrates color correction and defogging (outperforming DCP) and Peng et al.’s [19] model that incorporates depth estimation to improve image quality. Third, there are innovative approaches: Zhuang et al.’s [20] Super-Laplacian algorithm for color restoration, Li et al.’s [21] method of splitting high/low-frequency components for enhancement, Yan et al.’s [22] bio-normalization model simulating biological vision to correct color bias, and Acharya et al.’s [23] histogram-splitting method to preserve image information. However, these methods heavily depend on prior assumptions, leading to insufficient robustness in complex underwater environments, with DCP prone to distortion in specific areas.

### 2.2. Underwater Image Enhancement Methods Based on Non-Physical Models

Non-physical model-based underwater image enhancement methods are mainly divided into two categories: spatial domain methods and transform domain methods. Spatial domain methods focus on directly manipulating image pixels, achieving enhancement by adjusting pixel gray values and optimizing local or global pixel distribution. The fusion-based method proposed by Ancuti et al. [24] uses dual-branch processing to obtain contrast enhancement results and color correction results, respectively, improving contrast while enhancing detail edges; Tolie et al.’s work [25] centers on addressing the poor quality of underwater images caused by light attenuation, water turbidity, and optical device limitations, introducing a blind quality assessment method that leverages channel-based structural features, chrominance dispersion rate scores, and overall saturation and hue and fuses these features through a multiple linear regression model, achieving superior accuracy, consistency, and computational efficiency in assessing both raw and enhanced underwater images; Muniraj et al. [26] focus on the principle of color constancy, introducing gamma correction to enhance color intensity and combining it with a defogging algorithm to design a correction factor, achieving a breakthrough in improving color cast and clarity; and the graph signal processing method by Sharma et al. [27] is more innovative, replacing traditional transformation methods with graph Fourier transform and image wavelet filter banks, effectively improving image clarity and contrast.

The core logic of transform domain methods is to map an image from the spatial domain to a specific transform domain and use the characteristics of this domain to achieve enhancement, with discrete cosine transform (DCT) being the current core technical direction. The total JND contour model based on DCT proposed by Sung-Ho Bae et al. [28] integrates multiple masking effects such as spatial contrast sensitivity function and luminance masking, indirectly optimizing image quality while improving image and video compression performance; the review study by Wan Azani Mustafa et al. [29] systematically sorts out the advantages and disadvantages of DCT and Discrete Wavelet Transform (DWT) methods, providing a theoretical reference for the development of subsequent image enhancement technologies; for specific scenarios, An [30] proposes a DCT-based local contrast enhancement detection algorithm, which enhances the AC coefficients of the target area in Synthetic Aperture Radar (SAR) images and exhibits better detection results and accuracy than traditional methods in complex noise environments; and the AIIE method proposed by Ju et al. [31] in the same period is more targeted, specifically adapted to scenarios with low-frequency interference such as underwater environments. It utilizes the statistical characteristics in the DCT domain and a designed mask to suppress low-frequency information and highlight high-frequency information, significantly improving image visibility. Overall, although non-physical model-based methods offer fast processing speeds and do not require explicit modeling, they often suffer from incomplete color cast correction and limited detail recovery. Moreover, by neglecting the underwater optical degradation process, these methods are prone to over- or underenhancement, exhibit poor adaptability in complex environments, lack robustness in enhancement performance, and may even introduce noise or artifacts.

### 2.3. Underwater Image Enhancement Method Based on Deep Learning

Underwater image enhancement technology based on deep learning acquires the mapping relationship between degraded images and reference images through end-to-end learning, breaking through the limitations of traditional physical models. Existing research mainly focuses on two directions: first, network innovation. For example, Md Jahidul Islam et al. [32] proposed a residual generative model that can handle both image enhancement and super-resolution tasks; Junaed Sattar et al. [33] developed the FUnIE-GAN network and constructed the EUVP dataset; Ankita Naik et al. [34] designed the lightweight network Shallow-UWnet; Restormer, proposed by Zamir et al. [35], can efficiently capture long-distance pixel information; the multi-color encoder by Li et al. [36] can solve the problems of low contrast and color spots in underwater images; Deep-Wave-Net, proposed by Preethi et al. [37], has a faster convergence speed; the DICAM model by Tolie et al. [38] can correct image color bias and distance-related degradation; and the Swin Transformer by Liu et al. [39] can adapt to multiple image tasks. Second, there is dataset improvement: Li et al. [40] constructed the UIEB dataset and proposed the Water_Net network, but this network failed to effectively solve the backscattering problem; Peng et al. [41] constructed the LSUI dataset, which contains richer scenes. However, these methods entail considerable computational overhead and necessitate high-quality paired training data. In complex environments, they frequently suffer from color deviations and over-smoothed details, and their generalization performance across diverse water types remains limited.

## 3. Our Method

The flowchart of our method is shown in Figure 1. A color transfer image is first generated by aligning the statistical distributions of the degraded input and a reference image and then fused with the original frame and an attenuation weight map to yield an initial color-corrected image; this corrected image is subsequently segmented through mean–median fusion and transformed by region-adaptive Sigmoid nonlinear mapping to produce a contrast-enhanced representation. In parallel, a grayscale version of the enhanced image is derived and a visual saliency map is computed via multi-scale feature extraction followed by adaptive smoothing, which serves as the dominant spatial weight for frequency domain gain control, while the enhanced image in YUV space undergoes block-wise DCT processing on the luminance (Y) channel with saliency-weighted gain adjustment and blocking artifact suppression through overlapping block weighted fusion; the refined Y channel is finally recombined with the U and V channels and converted back to the RGB space to deliver the detail-preserving enhanced result.

### 3.1. Color Correction

This paper proposes an improved CTCS [42] method. First, the in situ underwater degraded image and a reference image—selected from a well-illuminated natural dataset exhibiting standard color statistics and requiring no adaptive adjustment—are both transformed into the CIELab space. The three-channel statistical moments of each image are then computed, and channel-wise color matching is applied to the degraded image to generate the color-transferred result in the CIELab space. Subsequently, a dual-channel attenuation–interaction model is employed to derive an attenuation weight map, which is then fused with the color-transferred image and the original frame to yield the final color-corrected result. See Table A1 for the meanings of relevant symbols.

In the CIELab space, the mean and standard deviation of the original image and the reference image are calculated, respectively:(1)$$\begin{array}{c} \mu_{Lab}^{S}=\frac{1}{H}\sum_{x=1}^{H}\;I_{Lab}^{S}(x),\sigma_{Lab}^{S}={\left(\frac{1}{H}\sum_{x=1}^{H}{\left(I_{Lab}^{S}(x)-\mu_{Lab}^{S}\right)}^{2}\right)}^{\frac{1}{2}}\\ \mu_{Lab}^{R}=\frac{1}{H}\sum_{x=1}^{H}\;I_{Lab}^{R}(x),\sigma_{Lab}^{R}={\left(\frac{1}{H}\sum_{x=1}^{H}{\left(I_{Lab}^{R}(x)-\mu_{Lab}^{R}\right)}^{2}\right)}^{\frac{1}{2}} \end{array}$$

Here, $I_{Lab}^{S}(x)$ and $I_{Lab}^{R}(x)$ denote the values of the original image and the reference image at pixel x in the CIELab space; $\mu_{Lab}^{S}$ and $\sigma_{Lab}^{S}$ are the mean and standard deviation of the original image in the CIELab color space, while $\mu_{Lab}^{R}$ and $\sigma_{Lab}^{R}$ are the mean and standard deviation of the reference image in the CIELab color space; H=M×N is the total number of pixels. Based on these statistical features, we perform color transfer using the following formula:(2)$$I_{CT,Lab}(i)=\left(\frac{I_{Lab}(i)-\mu_{Lab}^{S}}{\sigma_{Lab}^{S}}\right)\cdot \sigma_{Lab}^{R}+\mu_{Lab}^{R}$$

Here, $I_{CT,Lab}$ is the result after color style transfer; $I_{Lab}(i)$ is the pixel value of the original degraded image in the i channel of the CIELab space (where i corresponds to the L, a, and b channels). After obtaining $I_{CT,Lab}$, the final corrected image $I_{\mathrm{CT}}$ in the RGB space can be obtained through the standard color space conversion from CIELab to RGB.

To mitigate the limitation of conventional color transfer methods in neglecting wavelength-dependent underwater attenuation, the algorithm first estimates channel-specific attenuation indices for the red and blue bands of the original image as(3)$$A_{r}=1-{\left(\frac{r}{255}\right)}^{\gamma_{1}}$$(4)$$A_{b}=1-{\left(\frac{b}{255}\right)}^{\gamma_{2}}$$

Among them, r and b are the normalized value of the red and blue channel individually, and $\gamma_{1}$ is the red light attenuation sensitivity parameter. In strong light scenes, it takes $\gamma_{1}=1.1$; in weak light scenes, it takes $\gamma_{1}=1.4$. $\gamma_{2}$ is the blue light scattering sensitivity parameter. In strong light scenes, $\gamma_{2}=0.9$ is taken, and in weak light scenes, $\gamma_{2}=0.6$ is taken. The attenuation weight map is generated:(5)$$W=\frac{A_{r}}{A_{r}+A_{b}+\epsilon }$$

Among them, ϵ=0.001 prevents division by zero errors; $W\in \left[0,1\right]$. The larger the value, the more the red light attenuation dominates the distortion in the region.

Ultimately, the corrected image is obtained through a weighted fusion of the attenuation map and the color transfer result:(6)$$I_{CC}\left(i,j\right)=W\left(i,j\right)\cdot I_{CT}\left(i,j\right)+\left(1-W\left(i,j\right)\right)\cdot I_{S}\left(i,j\right)$$

Here, $I_{CC}\left(i,j\right)$ denotes the final corrected image, $W\left(i,j\right)$ represents the attenuation weight map, $I_{CT}\left(i,j\right)$ is the color-transferred image, and $I_{S}\left(i,j\right)$ is the original degraded image. The proposed fusion scheme adaptively modulates the contribution of the color transfer term versus the original pixel values as a function of the local attenuation weight. In regions where attenuation is severe, the color transfer result dominates, thereby replenishing lost chromatic information; conversely, in weakly attenuated regions, the original data are preferentially preserved to safeguard natural appearance. Consequently, the model delivers accurate, location-aware correction of underwater color distortions while remaining consistent with human visual system characteristics and the physical propagation laws of light in water.

### 3.2. Contrast Enhancement

Following color correction, chromatic aberrations are notably reduced; however, the image still exhibits low global contrast arising from water column scattering and wavelength-dependent attenuation. Abdul Ghani et al. [43] used the mean as the segmentation threshold to divide the histogram into two sub-regions. Owing to the abundance of extreme gray-level values in underwater imagery, the sample median offers higher robustness against outliers than the arithmetic mean. Consequently, we define the optimal separation intensity $I_{sep,c}$ as the midpoint between the channel-wise mean and median, ensuring a stable partition of the tonal distribution.

With $I_{sep,c}$ serving as the channel-specific boundary, each histogram is partitioned into background and foreground sub-histograms. Both segments are then subjected to Sigmoid-based nonlinear stretching, where the compressive–expansive response expands low-contrast intervals while compressing high-contrast extremes, yielding a perceptually natural contrast enhancement that respects the original tonal distribution of underwater scenes.

The background region intensity mapping formula is(7)$$I_{B}=I_{min,c}+\left(I_{sep,c}-I_{min,c}\right)\times Sigmoid\left(P_{B}\right)$$
where $I_{B}$ is the intensity of a background pixel after stretching, $I_{min,c}$ is the minimum pixel intensity of channel c, and $P_{B}$ is the cumulative distribution probability of the background pixel.

The foreground region intensity mapping formula is(8)$$I_{F}=\left(I_{sep,c}+1\right)+\left(I_{max,c}-\left(I_{sep,c}+1\right)\right)\times Sigmoid\left(P_{F}\right)$$
where $I_{F}$ is the intensity of a foreground pixel after stretching, $I_{max,c}$ is the maximum pixel intensity of channel c, and $P_{F}$ is the cumulative distribution probability of the foreground pixel.

Upon completion of the channel-wise regional Sigmoid stretching, the background sub-images $I_{B}$ of all channels are merged to yield a low-enhancement image that preserves dark-region details while maintaining subdued foreground brightness. Conversely, the foreground sub-images $I_{F}$ are integrated to produce an overenhancement image that accentuates bright-area details at the cost of amplifying background noise. The two complementary images are finally combined to generate the contrast-balanced enhanced result $I_{BF}$.

### 3.3. Detail Enhancement

Two complementary operations are executed in parallel on the contrast-enhanced image. First, the image is converted to grayscale, and a visual saliency map is computed via multi-scale feature extraction followed by adaptive smoothing; this map subsequently serves as the primary spatial weight for frequency domain gain control. Second, the enhanced image is transformed into the YUV space, where the luminance (Y) channel undergoes block-wise discrete cosine transform (DCT) enhancement guided by the saliency weights. Blocking artifacts are mitigated through overlapping-block weighted fusion. Finally, the refined Y channel is recombined with the U and V channels, and the composite image is converted back to the RGB color space to yield an enhanced result with preserved detail clarity.

#### 3.3.1. Theoretical Basis

The human visual system exhibits heightened sensitivity to mid-frequency and high-frequency spatial components such as edges and fine-scale textures and spontaneously allocates selective attention to visually salient regions [44]. The discrete cosine transform (DCT)—a canonical frequency domain operator—orthogonally decomposes an N × N image block into a compact ensemble of cosine basis functions, thereby furnishing an analytical substrate for targeted spectral manipulation. Formally, the DCT transformation is defined as(9)$$D\left(u,v\right)=\alpha \left(u\right)\alpha \left(v\right)\sum_{x=0}^{N-1}\sum_{y=0}^{N-1}\left(x,y\right)cos\left(\frac{\left(2x+1\right)u\pi }{2N}\right)cos\left(\frac{\left(2y+1\right)v\pi }{2N}\right)$$

In the formula, u,v=0,1,…,N−1, $\alpha \left(u\right)=\sqrt{1}/N$ (u=0) or $\sqrt{2}/N$ (u>0). The same applies to $\alpha \left(v\right)$.

In the discrete cosine transform (DCT) domain, the coefficient located at position (0,0), known as the DC coefficient, represents the average intensity or luminance of the corresponding image block. The remaining coefficients, termed AC coefficients, encapsulate the spatial frequency details: high-frequency AC coefficients primarily characterize edges and textures, whereas low-frequency AC coefficients correspond to smoother regions within the block. This clear separation of information in the frequency domain offers a precise and manipulable basis for targeted image detail enhancement.

#### 3.3.2. Visual Salience Perception

In underwater imagery, the perceptual significance of different regions varies considerably. Applying uniform enhancement across the entire image may inadvertently amplify background noise or result in over-sharpening of non-critical areas. This approach fails to align with the way the human visual system (HVS) naturally prioritizes focal regions. To address this, we propose the integration of a visual saliency perception stage that emulates the HVS’s ability to identify key areas of interest, thereby providing regional weighting cues for subsequent adaptive enhancement [45].

To extract salient texture and edge information, we perform multi-scale Gabor filtering, expressed as(10)$$G_{\lambda ,\theta ,\psi ,\sigma ,\gamma }\left(x,y\right)=exp\left(-\frac{x^{\prime 2}+\gamma^{2}y^{\prime 2}}{2\sigma^{2}}\right)cos\left(2\pi \frac{x^{\prime }}{\lambda }+\psi \right)$$

In the formula, $x^{\prime }=xcos\theta +ysin\theta$, $y^{\prime }=-xsin\theta +ycos\theta$, λ, θ, ψ, σ, and γ represent the wavelength, direction, phase, standard deviation of the Gaussian function, and aspect ratio, respectively. Two Gabor filters in the horizontal direction ($\theta =0^{\circ },\lambda =0.15$) and the vertical direction ($\theta =90^{\circ },\lambda =0.2$) are selected to filter the luminance channel. The filtering results are denoted as $G_{1}\left(x,y\right)$ and $G_{2}\left(x,y\right)$. The preliminary saliency map $S\left(x,y\right)$ is obtained by adding the absolute values of the filtering results, as shown in Equation (11), to highlight the salient regions.(11)$$S\left(x,y\right)=\left|G_{1}\left(x,y\right)\right|+\left|G_{2}\left(x,y\right)\right|$$

To suppress noise and smooth the saliency map, adaptive normalization processing is performed: mapping S(x,y) to the [0, 255] interval to obtain $S_{norm1}\left(x,y\right)$ and then applying a Gaussian kernel $G_{gauss}\left(x,y\right)$ with a size of 21 × 21 and a standard deviation of 7 for blurring. A 21 × 21 kernel is large enough to cover the local neighborhood of salient regions, merging adjacent pixel features for effective noise suppression without excessively blurring salient details, while a standard deviation of 7 provides an appropriate blur strength. It is expressed as(12)$$S_{blur}\left(x,y\right)=S_{norm1}\left(x,y\right)*G_{gauss}\left(x,y\right)$$

The final result is normalized to the [0, 1] interval to obtain the final visual saliency map $S_{\mathrm{final}}\left(x,y\right)$ that reflects the regional attention, with higher values indicating that the region is more likely to attract visual attention.

#### 3.3.3. Frequency Domain Adaptive Gain Adjustment

Conventional frequency domain enhancement methods typically apply a uniform gain, which unavoidably amplifies high-frequency noise, over-emphasizes structurally irrelevant regions, and deviates from human visual perception. The discrete cosine transform (DCT) mitigates these drawbacks by decomposing the image into spectrally disjoint coefficients: low-frequency terms represent smooth areas, whereas high-frequency terms encode edges and textures. This separability enables tiered processing—the selective amplification of high-frequency components for detail sharpening while mildly adjusting low-frequency components for naturalness preservation—thereby achieving a perceptually convincing balance between clarity and fidelity. To address this, we propose a “three-level gain tuning” method that fuses DCT frequency characteristics with visual saliency to deliver precise, detail-orientated enhancement. Firstly, the luminance channel is subjected to block DCT transformation: the luminance channel $Y\left(x,y\right)$ is divided into 8 × 8 sub-blocks. For each sub-block $B\left(x,y\right)$, a DCT transformation is performed to obtain the coefficient matrix $D\left(u,v\right)$, and the intra-block AC energy is calculated to measure the regional contrast. The AC energy is defined as the sum of the squares of all coefficients within the block minus the square of the DC coefficient, that is,(13)$$E_{\mathrm{ac}}=\sum_{u=0}^{7}\sum_{v=0}^{7}\left(u,v\right)-D^{2}\left(0,0\right)$$

To suppress numerical fluctuations, the activation energy $E_{\mathrm{ac}}$ is first log-transformed and subsequently mapped to the range [0, 1] via the hyperbolic tangent function, yielding the normalized contrast measure C.

The frequency gain function is(14)$$G_{freq}\left(u,v\right)=\alpha_{max}\cdot C\cdot exp\left(-\beta \cdot f{\left(u,v\right)}^{\kappa }\right)$$

Among them, $\alpha_{\max}=1.2$ is the maximum gain coefficient, C is the contrast measurement value, β=0.12 controls the gain bandwidth, and κ=0.8 adjusts the gain rate to ensure that the high-frequency region achieves stronger enhancement. The gain factor is further optimized by incorporating edge-direction characteristics. Determine the edge direction based on the horizontal and vertical low-frequency change information in the DCT coefficients: calculate the horizontal low-frequency change coefficient $d_{x}=\left|D\left(0,1\right)\right|$ and the vertical low-frequency change coefficient $d_{y}=\left|D\left(1,0\right)\right|$ and define the edge ratio:(15)$$z=d_{y}/\left(d_{x}+\epsilon \right)$$

Here, $\epsilon =10^{-6}$ is used to avoid division by zero.

Adjust the directional gain factor $G_{\mathrm{factor}}$ based on the r value: if z>2.0 (vertical edge), then $G_{factor}=1.4$ (j<i, to enhance the vertical high-frequency region) or 1.0 (j≥i, to prevent overenhancement). If z<0.5 (vertical edge), then $G_{factor}=1.5$ (j<i, enhance the vertical high-frequency region) or 1.0 (j≥i,); for isotropic regions (0.5≤z≤2.0), set $G_{factor}$ = 1.2. The total gain is ultimately obtained by integrating the visual saliency, frequency gain, and direction factor. Among them, $S_{final}$ is the average value of $S_{\mathrm{final}}\left(x,y\right)$ in each small block. The formula for the total gain is shown as(16)$$G\left(u,v\right)=S_{final}\times G_{factor}\times G_{freq}\left(u,v\right)$$

Adjust the DCT coefficients using the total gain, and the adjustment formula is(17)$$D^{\prime }\left(u,v\right)=D\left(u,v\right)\times \left(1+0.8tanh\left(G\left(u,v\right)\right)\right)$$

Among them, the DC coefficient $D^{\prime }\left(0,0\right)$ remains unchanged to maintain the overall brightness stability of the image. After the coefficient adjustment is completed, the inverse DCT transformation is performed on each sub-block to reconstruct the enhanced image block $B^{\prime }\left(x,y\right)$.

#### 3.3.4. Enhanced Post-Processing

During the block-based frequency domain enhancement process, the discontinuity at the boundaries between blocks can easily cause blocking artifacts [46], and the gain adjustment may lead to local brightness anomalies. Additionally, there is a risk of color distortion in the channel-merging stage. To address these issues, the post-processing of enhancement employs a multi-strategy collaborative optimization approach: to reduce the blocking artifacts caused by block processing, the algorithm adopts an overlapping block strategy and a weighted fusion mechanism. The block movement step size is set to half the block size (i.e., 4 pixels), and the enhanced sub-blocks obtained from the inverse DCT are weighted using a Hanning window $H\left(x,y\right)$, which is defined as(18)$$H\left(x,y\right)=0.5\left(1-cos\left(2\pi x/7\right)\right)\cdot 0.5\left(1-cos\left(2\pi y/7\right)\right)$$

Accumulate the weighted sub-blocks to the output luminance channel $E\left(x,y\right)$ and simultaneously record the sum of the weights as $H_{acc}\left(x,y\right)$, that is:$E\left(x,y\right)\leftarrow E\left(x,y\right)+B^{\prime }\left(x,y\right)\cdot H\left(x,y\right)$, $H_{acc}\left(x,y\right)\leftarrow H_{acc}\left(x,y\right)+H\left(x,y\right)$. After the enhancement is completed, the output brightness channel is normalized through $E\left(x,y\right)/H_{acc}\left(x,y\right)$ to limit the brightness value within the range $\left[0,255\right]$ to avoid overexposure or underexposure. Finally, the enhanced luminance channel $E\left(x,y\right)$ is recombined with the temporarily stored chrominance channels $U\left(x,y\right)$ and $V\left(x,y\right)$ to form a YUV image, which is then converted back to the RGB color space to obtain the final color image with enhanced details. By combining the inverse DCT transformation with a superimposed weighting strategy, it effectively balances detail enhancement and block effect suppression, ultimately achieving a high-quality enhancement effect with clear details in the significant regions and high visual comfort.

## 4. Experiment and Analysis

In this section, the proposed method is compared with existing techniques, including DCTE [47], UNTV [48], PCDE [49], PCFB [50], UDHTV [51], ZSRM [52], and WFAC [53]. Among the comparison algorithms, UDHTV is a physical model-based enhancement method, UNTV, PCDE, PCFB, and ZSRM are non-physical model-based enhancement methods, and WFAC and DCTE are frequency domain enhancement methods. By choosing these seven types of algorithms for comparison, we can comprehensively cover mainstream technical routes such as physical modeling, non-physical constraints, and frequency domain processing. Through the performance differences among different methods in color cast correction, defogging, and detail enhancement, the comprehensive advantages and applicable scenarios of the proposed method are highlighted. Three representative low-quality image datasets are selected for the experiments, namely, UIEB [54], EUVP [55], and LSUI [56]. Both the UIEB and EUVP datasets focus on the systematic perceptual research and analysis of underwater image enhancement methods. The UIEB dataset contains 950 real underwater scene images collected from the Internet, while the EUVP dataset includes over 12,000 paired underwater images and 8000 unpaired underwater images, providing rich test samples for the performance evaluation of underwater image enhancement algorithms. The LSUI dataset is a large-scale underwater image dataset containing 5004 pairs of images, covering diverse underwater scenes and providing strictly selected high-quality reference images, specifically designed for underwater image enhancement research.

### 4.1. Qualitative Assessment

To visually verify the comprehensive effect of the proposed underwater image enhancement method, this section reports qualitative comparison experiments based on the UIEB, EUVP, and LSUI datasets. For the experiments, we selected DCTE, UNTV, PCDE, PCFB, UDHTV, ZSRM, and WFAC as mainstream comparison methods. Through visual effect comparison, the performance of the proposed method in color correction, contrast enhancement, and detail enhancement is evaluated. The qualitative comparison results are shown in Figure 2, which integrates the enhancement effects of the three datasets: (a)–(c) are samples from the UIEB dataset, (d)–(f) are samples from the EUVP dataset, and (g)–(i) are samples from the LSUI dataset. Each dataset contains three typical underwater images. The experiments verify the enhancement stability of the method in different underwater scenarios through the comparison of multiple dataset samples.

As shown in Figure 2, DCTE delivers a flat enhancement with no standout dimension: colors remain dim and shallow, while edges and textures gain little clarity, so complex patterns in LSUI still look almost identical to the original. UNTV fails to remove color casts—greenish or bluish hues linger—and keeps underwater creatures and reefs blurry; its low contrast leaves the entire image gray and lifeless. PCDE clearly overenhances: it over-brightens the scene, producing whitish “washed-out” areas that destroy natural underwater tones; aggressive sharpening turns texture edges ragged and noisy, strong contrast blows highlights and crushes shadows, flattening depth. PCFB tinges regions yellow, yielding a dark, disharmonious palette; weak detail recovery leaves edges soft, and blown contrast erases highlight information, breaks layering, and produces a harsh glare. The UDHTV algorithm has an obvious overenhancement problem. In terms of color correction, some areas show over-saturation, and the blue tone of scenes in the UIEB database is overly intensified, appearing unnatural. In terms of contrast enhancement, local highlight areas are overexposed, and the transition between dark and bright areas is harsh, which destroys the layering of the image. Although there is a certain effect in detail enhancement, overenhancement causes texture details to be masked by noise, resulting in relatively low overall visual comfort. ZSRM offers moderate enhancement—no obvious color cast—but its mild contrast still muddies light–dark separation; detail gains are limited, complex textures stay blurred, and edge acuity falls short of DCT. Finally, WFAC remains conservative, giving a hazy impression: blue-green tones are subdued, overall grayness is high, layering is weak, and the soft edge profile provides the lowest detail definition of all. Our column in Figure 2 clearly shows that the proposed method has the best comprehensive performance of the three core indicators. In color correction, it can effectively correct the color shift in underwater scenes, with no color bias or over-saturation, and is close to the true color of the scene. In contrast enhancement, by adaptively adjusting the gain of high-frequency coefficients, it enhances the overall contrast while avoiding local overexposure—details in dark areas are clearly presented, and bright areas are not over-brightened, significantly enhancing the image layering and visual impact. The detail enhancement effect is particularly outstanding: the directional adjustment module specifically strengthens edges and textures, making edges sharper and details richer without noise amplification, achieving a good balance between detail clarity and overall naturalness.

By comparing the 8 groups of results from 9 samples across the three datasets in Figure 2, the following conclusions can be drawn: UDHTV causes image distortion due to overenhancement, resulting in low visual comfort; WFAC adopts a conservative enhancement approach, suffering from insufficient enhancement, which leads to hazy images and missing details; ZSRM achieves moderate enhancement but performs poorly in detail processing, with no outstanding advantages in any dimension; DCTE shows a plain overall enhancement effect, featuring dark colors and limited detail improvement, thus performing mediocrely; UNTV has average color correction and insufficient detail and contrast enhancement, leading to high image grayscale and weak visual impact; PCDE’s overenhancement results in bright colors and whitish images, accompanied by noise amplification and damaged image layers; PCFB has problems such as insufficient details, partial yellowish tones and overexposed contrast, leading to uncoordinated color tones and low detail recognition; and the DCT algorithm achieves a better balance in color naturalness, contrast balance, and detail clarity, and its comprehensive performance is significantly superior to other algorithms.

### 4.2. Quantitative Evaluation

To quantitatively verify the performance of the method, in this section, seven metrics including UIQM [57] (Underwater Image Quality Measure), SSIM [58] (Structural Similarity Index), PSNR [59] (Peak Signal-to-Noise Ratio), UCIQE [57] (Underwater Color Image Quality Evaluator), information entropy [60] (IE), average gradient [61] (AG), and standard deviation [62] (SD) are adopted for evaluation with the UIEB, EUVP, and LSUI datasets. The quantitative results are summarized in Table 1, Table 2 and Table 3.

From the results of the UIEB dataset in Table 1, the DCT algorithm stands out in several key indicators. In terms of overall quality, its UIQM reaches 4.5542, significantly higher than other algorithms. Meanwhile, its UCIQE ranks first at 0.5100, highlighting its advantages in color balance and contrast. In terms of detail enhancement, it achieves the highest information entropy of 7.6433, an average gradient of 122.4165 that far exceeds similar algorithms, and a standard deviation of 67.5269 that leads by a large margin. These indicate its remarkable effects in detail preservation, edge clarity and contrast improvement. In terms of structural consistency, the SSIM of DCT (0.8172) is slightly lower than that of PCFB (0.8270) and DCTE (0.8211), but it still remains at a good level. Its PSNR (31.3165) is close to that of most algorithms, belonging to the upper-middle level. In summary, although the DCT algorithm is not the best in terms of SSIM and PSNR, it has obvious advantages in core indicators such as UCIQE and information entropy. Its overall performance is stable and excellent, especially in color balance, detail preservation, and contrast improvement.

The results for the EUVP underwater image dataset in Table 2 show that the DCT algorithm has significant advantages. In terms of overall quality, its UIQM reaches 4.9359, which is significantly higher than that of other algorithms; its UCIQE stands at 0.4915, second only to ZSRM (0.4942), still highlighting its advantages in color balance and contrast. In terms of detail enhancement, it achieves the highest information entropy of 7.7333, an average gradient of 146.1495 that far exceeds similar algorithms (with WFAC ranking second at 125.6689), and a standard deviation of 71.7036 that leads by a large margin, which indicates its remarkable effects in detail preservation, edge clarity, and contrast improvement. In terms of structural consistency, the SSIM of DCT (0.8093) is slightly lower than that of DCTE (0.8148) and UDHTV (0.8129) but higher than most algorithms such as UNTV (0.7748) and ZSRM (0.7340), still remaining at a good level; its PSNR (31.4466) is close to the values of DCTE (31.4721), UNTV (31.4732) and other algorithms, belonging to the upper-middle level. In summary, although the DCT algorithm is not the best in terms of SSIM and PSNR, it has obvious advantages in core indicators such as UIQM, information entropy, average gradient, and standard deviation, with stable and excellent overall performance, especially in color balance, detail preservation, and contrast improvement.

The results for the LSUI dataset in Table 3 further validate the advantages of the DCT algorithm. In terms of overall quality, its UIQM reaches 4.5252, significantly higher than other algorithms (with WFAC ranking second at 3.9052); its UCIQE ranks first at 0.4883, far exceeding other algorithms, highlighting its advantages in color balance and contrast. In terms of detail enhancement, its IE (information entropy) is 7.6418, second only to WFAC (7.7409); its AG (average gradient) is 146.0433, which far surpasses similar algorithms; and its SD (standard deviation) leads by a large margin at 69.2745, indicating its remarkable effects in detail preservation, edge clarity, and contrast improvement. In terms of structural consistency, the SSIM of DCT (0.7783) is slightly lower than that of UNTV (0.8056) and UDHTV (0.7998) but higher than that of ZSRM (0.7720) and WFAC (0.6870), still remaining at a reasonable level; its PSNR (31.3688) is close to the values of ZSRM (31.4130), UDHTV (31.3572), and other algorithms, belonging to the upper-middle level. In summary, although the DCT algorithm is not the best in terms of SSIM and PSNR, it has obvious advantages in core indicators such as UIQM, UCIQE, average gradient, and standard deviation, with stable and excellent overall performance, especially in color balance, detail preservation, and contrast improvement.

Across the three benchmark datasets, the proposed DCT method consistently ranks first in terms of global visual quality, detail fidelity, and contrast enhancement. Its superior edge-sharpening and texture-preserving capabilities directly address the core degradations encountered in underwater imaging. Among the competitors, DCTE attains marginally higher scores on certain structural consistency indices, whereas WFAC occasionally surpasses others on isolated detail metrics; UNTV, PCDE, and PCFB exhibit intermediate performance. Overall, none of the alternative algorithms match the comprehensive enhancement delivered by the DCT framework.

### 4.3. Ablation Experiment

To quantitatively evaluate the contribution of each constituent module—color correction, contrast enhancement, and DCT-based detail refinement—systematic ablation studies were conducted on the UIEB dataset. Progressive removal of individual components and subsequent performance comparison elucidate both their standalone efficacy and their synergistic role in improving underwater image quality.

For the ablation experiment, we selected representative samples from the UIEB dataset and set up four comparison scenarios to isolate the influence of each component: the original image; the method without color correction (-w/o Color Correction, -w/o CC): removing the color correction module while retaining the contrast enhancement and DCT detail enhancement processes; the method without contrast enhancement (-w/o Contrast Enhancement, -w/o CE): removing the contrast enhancement module while retaining the color correction and DCT detail enhancement processes; and the method without DCT detail enhancement (-w/o DCT Detail Enhancement, -w/o HDE): removing the DCT detail enhancement module while retaining the color correction and contrast enhancement processes. The experiment adopted a combined qualitative and quantitative evaluation approach: qualitative results are presented through visual comparison images, as shown in Figure 3, to demonstrate the visual effect differences among various scenarios; quantitative assessment was conducted by calculating metrics such as UIQM (Underwater Image Quality Measure), SSIM (Structural Similarity Index), PSNR (Peak Signal-to-Noise Ratio), UCIQE (Underwater Color Image Quality Evaluation), information entropy (IE), average gradient (AG), and standard deviation (SD). The average scores of each scenario on the UIEB dataset are summarized in Table 4, with bolded values indicating the best results for the corresponding metrics.

Figure 3 shows the original underwater images selected from the UIEB dataset, as well as the enhancement results of the complete method and four ablation scenarios. Visual inspection unambiguously reveals the function of each module. In the absence of color correction (-w/o CC), the image retains moderate contrast and coarse details yet suffers from a pronounced blue-green cast caused by wavelength-selective attenuation; the foreground and background exhibit poor chromatic coherence, yielding an unnaturally rigid appearance. This confirms that the color correction module is essential for neutralizing scattering-induced color shift and for re-balancing channel intensities. When contrast enhancement is disabled (-w/o CE), the global histogram is restored to a natural palette, but the dynamic range collapses: dark-region details are submerged, bright-region highlights are compressed, and the entire image appears veiled and “grayish”. Even though the detail enhancement module is active, the low-contrast baseline prevents texture information from being perceived, underscoring the role of contrast enhancement in expanding the dynamic range. Without DCT detail enhancement (-w/o HDE), color fidelity and global contrast remain satisfactory, yet object boundaries and fine textures are blurred, and the overall plasticity is reduced; in complex regions, high-frequency details are clearly lost. This demonstrates that the DCT module, by boosting high-frequency coefficients in a visually adaptive manner, compensates for the local information deficit that contrast enhancement alone cannot restore.

Table 4 presents the average quantitative evaluation results for the complete method and three ablation scenarios on the UIEB dataset. Bold entries denote the best result for each metric. Ablation results demonstrate that the full algorithm consistently outperforms all degraded variants, corroborating the synergistic value of color correction, contrast enhancement, and DCT-based detail refinement. Removing color correction (-w/o CC) reduces UIQM to 2.7586 and UCIQE to 0.4100, confirming that this module is indispensable for natural color recovery and overall perceptual quality. Eliminating contrast enhancement (-w/o CE) decreases PSNR to 22.28 dB; although the average gradient remains high (101.11), the drop in luminance fidelity and global quality underscores the module’s role in boosting image layering. Excluding DCT detail enhancement (-w/o HDE) lowers the average gradient to 91.79 and information entropy to 7.583, revealing a clear loss in texture acuity and fine-detail richness. The complete algorithm, integrating all three modules, attains UIQM = 4.5542, PSNR = 31.32 dB, UCIQE = 0.5100, and average gradient = 122.42—delivering optimal color naturalness, contrast stratification, and detail sharpness.

### 4.4. Computational Complexity Experiment

#### 4.4.1. Experimental Set Up

The hardware environment of this experiment adopts an Intel Core i7-12700H CPU, an NVIDIA RTX 3060 GPU, and 32 GB DDR4 3200 MHz memory. The Intel Core i7-12700H CPU is manufactured by Intel Corporation, with its source city being Santa Clara, CA, USA. The graphics card selected for the experiment is the NVIDIA RTX 3060 GPU, which is produced by NVIDIA Corporation and also sourced from Santa Clara, CA, USA. The memory configuration is 32 GB DDR4 3200 MHz, adopting products of the Samsung brand, whose source city is Suwon, Republic of Korea. The software environment is built based on the Windows 11 Professional operating system, using Python 3.9 as the programming language and relying on OpenCV 4.8.0 and NumPy 1.25.2 for image processing and numerical calculation. The runtime measurement is implemented through the timeit. Timer tool, and each group of tests is repeated 50 times to eliminate errors caused by random system fluctuations. For dataset selection, 100 representative samples are randomly selected from each of the three benchmark datasets (UIEB, EUVP, and LSUI), and all samples are uniformly adjusted to a resolution of 640 × 480 to ensure a consistent data scale. The measurement scope of the color correction component starts from CIELab space conversion and ends at the fusion of the attenuation weight map and the color transfer image; the contrast enhancement component starts from the mean–median fusion threshold calculation and ends at the background/foreground image fusion; the detail enhancement component starts from Gabor filtering and ends at inverse DCT and overlapping block weighted fusion.

#### 4.4.2. Experimental Results and Analysis

Table 5 shows the average runtime of the three core components on the UIEB, EUVP, and LSUI datasets. As can be seen from Table 5, the runtime of each component follows a consistent pattern across the three datasets. The detail enhancement component has the longest average runtime, which is 16.87 ms, and it is the main source of the algorithm’s computational overhead; the color correction component ranks second; and the contrast enhancement component has the shortest runtime. At the same time, the runtime fluctuation of the same component across different datasets is small, with the maximum difference being less than 0.2 ms. Combined with the unified resolution and experimental parameter settings, this verifies the stability and reliability of the complexity results and also provides a clear data basis for the subsequent engineering optimization of the algorithm.

### 4.5. Application Testing

To verify the practical value of the enhancement method in real visual tasks, this section applies the enhanced images to two typical downstream tasks: edge detection and key point detection. By comparing the task results of the original images with those of the enhanced images, the promoting effect of the enhancement method on visual feature extraction is evaluated. The tests are based on representative samples from the EUVP dataset. Edge detection uses the Canny operator, and key point detection uses the SIFT algorithm. The effects are comprehensively analyzed through qualitative and quantitative indicators.

#### 4.5.1. Edge Detection

Edges are fundamental visual cues for characterizing image structure and object boundaries, and the fidelity of edge extraction is inherently determined by the magnitude of gray-level gradients [63]. In raw underwater images, contrast attenuation induces gradient smoothing, which causes the Canny edge detector to generate fragmented contours, missing weak edges, and spurious responses. As illustrated in Figure 4, compared with seven mainstream underwater image enhancement algorithms (i.e., DCTE, UNTV, PCDE, PCFB, UDHTV, ZSRM, and WFAC), the proposed algorithm effectively enhances edge intensity and enriches fine-grained detail information by targeted the amplification of gray-level gradients. Consequently, it achieves continuous object silhouettes, preserved weak edges, and improved edge-background separation, thereby significantly improving the accuracy of edge detection for underwater scenes.

#### 4.5.2. Key Point Detection

The key points serve as the basis for local feature matching, and their quantity and stability depend on the clarity of the detail texture. Due to the blurriness of the details in the original underwater images, the key points extracted by the SIFT algorithm are concentrated in high-contrast areas, with a sparse distribution and weak discrimination of neighborhood features. Through detail enhancement, the local features in low-contrast areas (such as shadow areas and weak-textured surfaces) of the enhanced images are highlighted. As shown in Figure 5, the number of key points significantly increases and their distribution becomes more uniform. The quantitative results in Table 6 show that the average number of key points in the enhanced images increases by 107%, verifying the effectiveness of the enhancement method in enriching and stabilizing local features.

#### 4.5.3. Object Detection

Object detection underpins critical underwater downstream tasks such as biological monitoring and search-and-rescue; its accuracy is contingent upon feature discriminability and contour sharpness. To quantify the detection benefit conferred by enhancement, we constructed a benchmark of 50 representative EUVP images containing fish, coral colonies, and man-made equipment and annotated them with axis-aligned bounding boxes in Pascal VOC format. A COCO-pre-trained YOLOv5s detector was fine-tuned on this subset while keeping the IoU threshold at 0.5. The original frames (baseline) and their enhanced counterparts—produced by DCTE, UNTV, PCDE, PCFB, UDHTV, ZSRM, WFAC, and the proposed DCT method—were fed into the identical network. The mean precision (P)24 [64], recall (R) [64], F1-score [65], and F1-score improvement across 50 independent runs are reported in Table 7. F1-score improvement refers to the relative increase in the F1-score achieved by each algorithm compared to the F1-score of the original image. This metric is used to quantify the algorithm’s enhancement in object detection performance relative to the original image. It is calculated using the following formula:(19)$$F1-score\;improvement=\frac{F1_{algorithm}-F1_{original}}{F1_{original}}\times 100\%$$

Table 7 confirms that raw underwater images yield the lowest scores, underscoring the detrimental effect of aquatic degradation on detection performance. Among the competing algorithms, DCTE ranks lowest; UNTV, PCFB, and PCDE deliver moderate results; and UDHTV, ZSRM, and WFAC perform comparatively better yet still exhibit deficiencies such as missing details, residual color casts, or overenhancement. In sharp contrast, the proposed DCT method achieves the highest precision, recall, and F1-score. Relative to WFAC, these metrics rise by 15.7%, 14.0%, and 14.8%, respectively, while the F1-score surpasses that of the original images by 51.7%. This substantial gain originates from the joint optimization of color correction, contrast enhancement, and detail boosting, which collectively suppress both false positives and false negatives. The pronounced margin over all competitors validates the practical utility of the DCT method for real-world underwater vision tasks.

As illustrated in Figure 6, this figure presents the qualitative comparison results of underwater object detection between the proposed algorithm and seven mainstream methods (i.e., DCTE, UNTV, PCDE, PCFB, UDHTV, ZSRM, and WFAC), where the detection bounding boxes of all algorithms are marked in red. From the perspective of the localization accuracy and coverage range of the bounding boxes, it can be observed that the detection boxes of other comparative algorithms generally exhibit an over-expansion phenomenon: the size of the bounding boxes is excessively large, failing to accurately fit the target contours. Some algorithms even include background regions within the bounding boxes or miss key local parts of the targets, which reflects their insufficient capability to identify underwater target boundaries. In contrast, the detection boxes of the proposed algorithm can strictly match the actual contours of the targets with optimal size adaptability, without obvious over-coverage or under-coverage issues, thus accurately capturing the complete morphology of underwater targets. This difference originates from the synergistic effect of multi-channel attenuation analysis and DCT-based detail enhancement in the proposed algorithm, which effectively improves the edge clarity of images and the separability between targets and backgrounds. Consequently, more reliable visual features are provided for the YOLOv5s detector, thereby significantly optimizing the localization accuracy of the detection boxes and achieving precise detection of underwater targets.

## 5. Conclusions

To tackle the three major challenges of underwater imaging—color cast, low contrast, and blurred details—this paper presents a unified enhancement framework that couples multi-channel attenuation analysis with DCT. By integrating physical priors with frequency domain processing, the proposed method breaks the limitation of conventional schemes that handle these issues in isolation. A three-level synergy of color correction, contrast stretching, and detail sharpening is designed as the core: each module has a dedicated role while mutually reinforcing the others, enabling parameter-free, adaptive enhancement. Experiments on three public datasets—UIEB, EUVP, and LSUI—as well as on downstream tasks such as object detection and edge extraction demonstrate that our results significantly outperform mainstream algorithms such as UDHTV in both visual quality and no-reference metrics like UIQM and UCIQE. Ablation studies further confirm that all three modules are indispensable. The algorithm respects human visual characteristics and underwater light propagation physics, effectively suppressing overenhancement and artifacts. Future work will focus on refining the attenuation model and developing lightweight implementations to facilitate real-time applications.

## Figures and Tables

**Figure 1 sensors-25-07192-f001:**
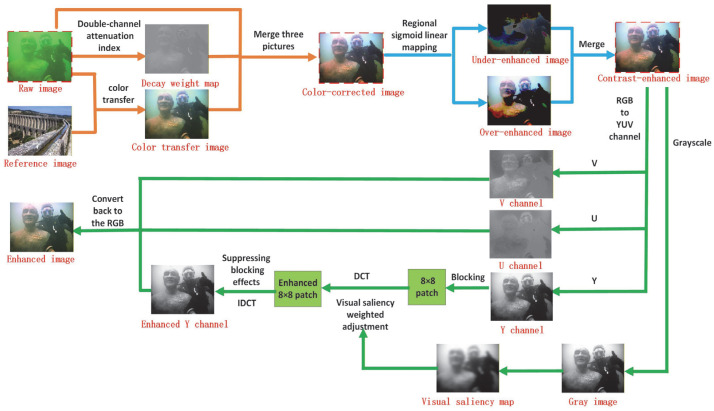
Flowchart of the algorithm.

**Figure 2 sensors-25-07192-f002:**
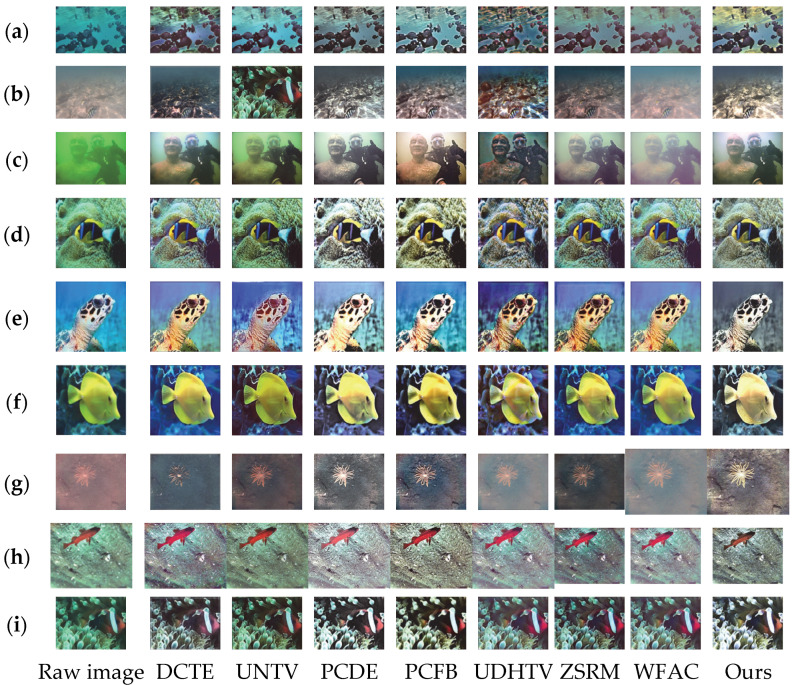
Comparison of the effects of different algorithms on three underwater image datasets. (**a**–**c**) are samples from the UIEB dataset, (**d**–**f**) are samples from the EUVP dataset, and (**g**–**i**) are samples from the LSUI dataset.

**Figure 3 sensors-25-07192-f003:**
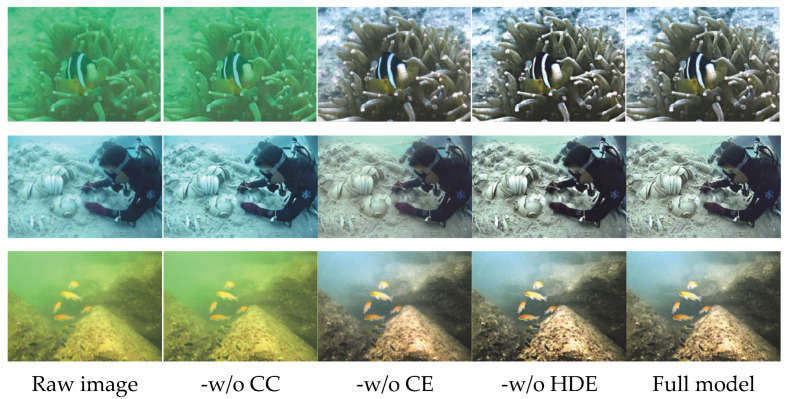
Qualitative comparison results of ablation experiments.

**Figure 4 sensors-25-07192-f004:**
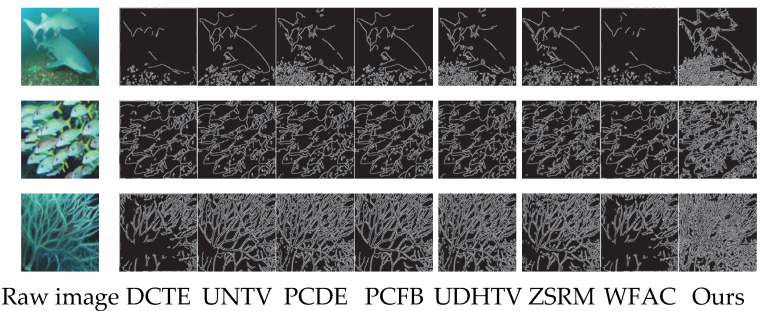
Qualitative comparison of edge detection results between the proposed algorithm and mainstream underwater image enhancement algorithms.

**Figure 5 sensors-25-07192-f005:**
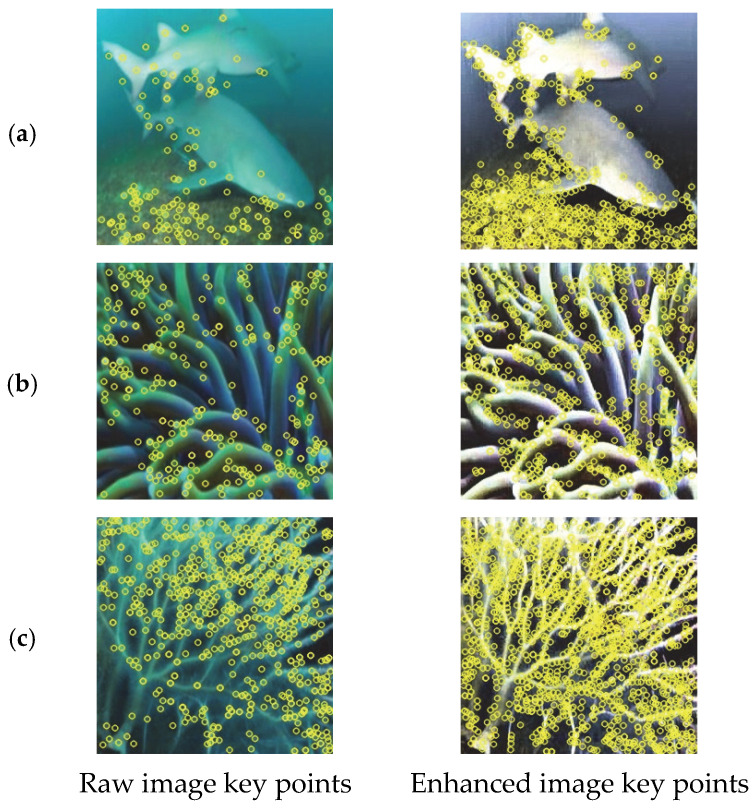
Comparison of key point distribution between the original image and the enhanced image. (**a**–**c**) respectively correspond to three representative groups of underwater image samples in the EUVP dataset. Each sub-figure pair shows the key point detection results of the original image and the enhanced image.

**Figure 6 sensors-25-07192-f006:**
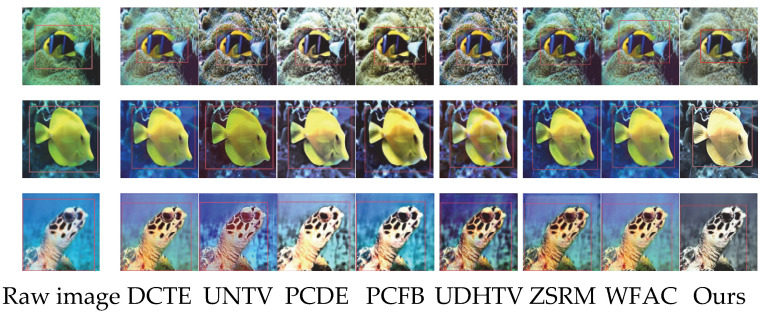
Qualitative comparison of underwater object detection results between the proposed algorithm and mainstream methods.

**Table 1 sensors-25-07192-t001:** Experimental comparison data of each algorithm for the UIEB dataset. All bold values in the table indicate the optimal results under the corresponding evaluation metrics, and italic values denote the suboptimal results.

	UIQM	SSIM	PSNR	UCIQE	IE	AG	SD
DCTE	4.0210	**0.8211**	*31.3231*	0.4620	7.0321	101.5100	40.4632
UNTV	4.1221	0.8112	31.3101	0.4576	7.1221	104.3201	45.6781
PCDE	4.3214	0.8138	31.3019	0.4621	7.1456	101.7892	40.7829
PCFB	4.4123	0.8270	31.3028	*0.4762*	7.4653	*113.3398*	44.7921
UDHTV	4.4022	0.8162	31.3131	0.4410	6.9961	101.4178	40.4631
ZSRM	**4.9806**	0.8074	31.1688	0.4744	6.9978	71.2831	40.4722
WFAC	3.8381	0.7820	**31.3575**	0.4178	*7.5633*	112.4023	*55.3321*
DCT	*4.5542*	*0.8172*	31.3165	**0.5100**	**7.6433**	**122.4165**	**67.5269**

**Table 2 sensors-25-07192-t002:** Experimental comparison data of each algorithm for the EUVP dataset. All bold values in the table indicate the optimal results under the corresponding evaluation metrics, and italic values denote the suboptimal results.

	UIQM	SSIM	PSNR	UCIQE	IE	AG	SD
DCTE	*4.6328*	**0.8148**	31.4721	0.4770	7.2403	101.1231	49.2145
UNTV	4.4892	0.7748	**31.4732**	0.4711	7.2431	102.1293	50.9128
PCDE	4.5011	0.7990	31.4001	0.4521	7.3129	102.4928	52.4938
PCFB	4.6122	0.8100	31.4529	0.4811	7.3190	103.4132	53.8732
UDHTV	4.4341	*0.8129*	*31.4685*	0.4774	7.2396	99.1021	48.1158
ZSRM	4.4036	0.7340	31.4019	**0.4942**	7.2315	102.3219	50.0777
WFAC	4.4506	0.6470	31.1022	0.4337	*7.6899*	*125.6689*	*59.5134*
DCT	**4.9359**	0.8093	31.4466	*0.4915*	**7.7333**	**146.1495**	**71.7036**

**Table 3 sensors-25-07192-t003:** Experimental comparison data of each algorithm for the LSUI dataset. All bold values in the table indicate the optimal results under the corresponding evaluation metrics, and italic values denote the suboptimal results.

	UIQM	SSIM	PSNR	UCIQE	IE	AG	SD
DCTE	3.3487	0.7912	31.3575	0.4366	7.0558	71.4901	43.2123
UNTV	3.6171	**0.8056**	31.3417	0.4482	7.2341	102.7821	43.8989
PCDE	3.3813	0.7978	31.3122	0.4511	7.3123	108.7891	44.5879
PCFB	3.8321	0.7928	31.3322	0.4611	7.4123	111.2342	44.9876
UDHTV	3.3630	*0.7998*	31.3572	0.4369	7.0558	71.4888	41.3137
ZSRM	3.8696	0.7720	**31.4130**	*0.4683*	7.2024	90.5919	45.9988
WFAC	*3.9052*	0.6870	31.2243	0.4306	**7.7409**	*121.1898*	*56.2596*
DCT	**4.5252**	0.7783	*31.3688*	**0.4883**	*7.6418*	**145.0433**	**69.2745**

**Table 4 sensors-25-07192-t004:** Quantitative results of ablation experiments based on the UIEB dataset.

	UIQM	SSIM	PSNR	UCIQE	IE	AG	SD
Raw image	1.9544	0.7639	19.4438	0.3221	6.8900	41.3818	36.3170
-w/o CC	2.7586	0.7822	21.4057	0.4100	7.0645	61.4828	41.5255
-w/o CE	4.1832	0.7911	22.2839	0.4511	7.1490	101.1127	72.4165
-w/o HDE	3.9916	0.7933	22.1154	0.4479	7.5827	91.7882	59.1186
Complete	4.5542	0.8122	31.3165	0.5100	7.6433	122.4165	67.5269

**Table 5 sensors-25-07192-t005:** Average runtime of each component on the three datasets.

	Color Correction Runtime (ms)	Contrast Enhancement Runtime (ms)	Detail Enhancement Runtime (ms)	Total Runtime (ms)
UIEB	8.21 ± 0.35	5.17 ± 0.22	16.83 ± 0.51	30.21
EUVP	8.35 ± 0.41	5.23 ± 0.25	17.02 ± 0.48	30.60
LSUI	8.18 ± 0.38	5.12 ± 0.23	16.75 ± 0.45	30.15
Average	8.25 ± 0.38	5.17 ± 0.23	16.87 ± 0.48	30.29

**Table 6 sensors-25-07192-t006:** Comparison of key point quantities. (a)–(c) correspond to the subfigures of Figure 5.

	Raw Image	Enhanced Image	Increase in Magnitude
(a)	707	1463	108%
(b)	304	677	122%
(c)	469	902	92%

**Table 7 sensors-25-07192-t007:** Comparison of target detection results of different algorithms. All bold values in the table indicate the optimal results under the corresponding evaluation metrics.

	P	R	F1	F1 Improvement
Raw image	0.628	0.541	0.582	-
DCTE	0.685	0.623	0.653	12.2%
UNTV	0.712	0.668	0.689	18.4%
PCDE	0.698	0.647	0.672	15.5%
PCFB	0.689	0.656	0.672	15.5%
UDHTV	0.775	0.700	0.735	26.3%
ZSRM	0.753	0.691	0.721	23.9%
WFAC	0.764	0.714	0.738	26.8%
DCT	**0.892**	**0.875**	**0.883**	**51.7%**

## Data Availability

Data are contained within the article.

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
