# Peer review of "DCT Underwater Image Enhancement Based on Attenuation Analysis"

_sensors, 2025, doi:10.3390/s25237192_

Round 1

Reviewer 1 Report

Comments and Suggestions for Authors

Comments for the authors after reviewing the paper:

1.In Section 2 (Related Work), the authors are advised to divide the content into three subsections for better structure and readability.

2.In Section 3, the authors employ YUV color space conversion. Why not use HSI or other color space conversions instead? Please explain the advantages of using YUV.

3.In Section 4, the comparison algorithms should be arranged according to the order of publication in the literature. Accordingly, the algorithms, experimental figures, and corresponding metric tables should be reorganized.

4.The number of comparison algorithms in the experiments is too small (only four). The authors should consider adding three to four more comparison algorithms.

5.How is the object detection performance on the enhanced images? Please provide relevant results or analysis.

6.For non-reference underwater images, it is recommended that the authors perform simulation verification and list the corresponding non-reference quality metrics.

7.Why did the authors not include comparative experiments with deep learning-based algorithms?

8.Regarding Figure 4, which shows edge detection results, please provide a high-quality reference paper to help understand the purpose and significance of edge detection.

9.In Section 4.2, concerning the PSNR metric, the authors may have overlooked the following reference:

[1] Li, L.; Song, S.; Lv, M.; Jia, Z.; Ma, H. Multi-Focus Image Fusion Based on Fractal Dimension and Parameter Adaptive Unit-Linking Dual-Channel PCNN in Curvelet Transform Domain. Fractal Fract. 2025, 9, 157. https://doi.org/10.3390/fractalfract9030157

For the SSIM metric, the following reference may also have been missed:

[1] Cao, Z.; Liang, Y.; Deng, L.; Vivone, G. An Efficient Image Fusion Network Exploiting Unifying Language and Mask Guidance. IEEE Transactions on Pattern Analysis and Machine Intelligence, 2025, 47(11): 9845–9862. DOI: 10.1109/TPAMI.2025.3591930

Reviewer 2 Report

Comments and Suggestions for Authors

The paper proposes an underwater image enhancement method by integrating the multi-channel attenuation analysis with the discrete cosine transform. The proposed method produces a colour corrected image by fusing the calculated colour transfer image, original, and the attenuation weight map followed by global contrast balance through nonlinear mapping of gray-scale range. The saliency map-based high-frequency region identification and adaptive gain is applied to enhance details. The methodology introduced in the paper is novel and validated through extensive experiments. Detailed comments are below:

  1. The abstracts need to be re-written to clarify the definition of colour transfer image, as well as the difference between the original and reference images. The authors need to clarify whether their method require access to pristine reference image (which is not usually available in underwater domain).
  2. The authors need to update the references with more recent publications from the past few years. Below are some suggestions:
  3. Tolie, H.F., Ren, J., Cai, J., Chen, R. and Zhao, H., 2025. Blind Quality Assessment Using Channel-Based Structural, Dispersion Rate Scores, and Overall Saturation and Hue for Underwater Images. IEEE Journal of Oceanic Engineering.
  4. In the colour correction module, the authors need to specify how a reference image is selected and whether this needs to be changed adaptively based on the underwater image being enhanced.
  5. The experimental results need to be expanded by taking into account the computational complexity of the various components of the proposed method.

Reviewer 3 Report

Comments and Suggestions for Authors

In this paper, authors proposed an underwater image enhancement method that integrates multi-channel attenuation analysis and discrete cosine transform (DCT). Authors did more experiments to show the performance of the proposed method.

There are some comments below:

  1. Paper presentation could be improved. For example, some common parts such as subsections 3.3.1 and 3.3.2 could be shortened. In addition, it is better to have a symbol notation table for improving the readability of this paper.
  2. It would be better to mention the contribution of the proposed method.
  3. Figure 1 shows the flowchart of the proposed method. It would be better to explain how to select the reference image.
  4. Figure 1 shows that DCT is adopted in the proposed method. It would be better to explain why DCT is selected in the proposed method.
  5. Some hyper-parameters are not described clearly. For example, why the size of Gaussian kernel is 21 × 21 and its standard deviation is 7?
  6. Figure 2 shows some resulting images. It would be better to do subjective evaluation.

Round 2

Reviewer 1 Report

Comments and Suggestions for Authors

After reviewing the revised manuscript, I find that the authors have addressed some of my previous concerns. However, the following issues still need to be further clarified or revised:

1.In Figure 2, the image enhancement results from the three different datasets should be presented, and it should be clearly indicated which images belong to the same dataset.

2.Regarding the purpose of edge detection in Figure 4, the authors have not yet answered my question. I would like to know which high-quality reference has performed edge detection on enhanced images as you did, and what the purpose of doing so is.

3.From the results, the effect of edge detection seems to have little significance. For the three evaluation metrics used in Section 4.5.3, the authors need to cite relevant references. For example, the F1-score can be referenced as follows: [1] Li, L.; Ma, H.; Zhang, X.; Zhao, X.; Lv, M.; Jia, Z. Synthetic Aperture Radar Image Change Detection Based on Principal Component Analysis and Two-Level Clustering. Remote Sens. 2024, 16, 1861. https://doi.org/10.3390/rs16111861

4.For the PSNR metric, the following reference may also be helpful: [1] Lv, M.; Song, S.; Jia, Z.; Li, L.; Ma, H. Multi-Focus Image Fusion Based on Dual-Channel Rybak Neural Network and Consistency Verification in NSCT Domain. Fractal Fract. 2025, 9, 432. https://doi.org/10.3390/fractalfract9070432

5.What does the “F1-SCORE improvement” in Table 7 refer to, and how is it calculated? Please provide an explanation for this.

6.In Section 4.5.3, could the authors also include some example images of the object detection experimental results?

Reviewer 3 Report

Comments and Suggestions for Authors

Authors had addressed the reviews' comments in this version.

Round 3

Reviewer 1 Report

Comments and Suggestions for Authors

After carefully reviewing the revised manuscript, the authors have addressed most of my concerns. However, there is one minor issue that remains unresolved: regarding the PSNR metric, the authors may have overlooked the following reference: Multi-Focus Image Fusion Based on Dual-Channel Rybak Neural Network and Consistency Verification in NSCT Domain. Fractal Fract. 2025, 9, 432. https://doi.org/10.3390/fractalfract9070432.
